# Evaluation of Disposal Stability for Cement Solidification of Lime Waste

**DOI:** 10.3390/ma15030872

**Published:** 2022-01-24

**Authors:** Jong-Sik Shon, Hyun-Kyu Lee, Gi-Yong Kim, Tack-Jin Kim, Byung-Gil Ahn

**Affiliations:** Radwaste Management Center, Korea Atomic Energy Research Institute, 111, Daedeok-daero 989beon-gil, Yuseong-gu, Daejeon 34057, Korea; njsshon@kaeri.re.kr (J.-S.S.); kgyong@kaeri.re.kr (G.-Y.K.); ktj@kaeri.re.kr (T.-J.K.); bgan@kaeri.re.kr (B.-G.A.)

**Keywords:** AUC, lime waste, WAC, cementation, operation conditions

## Abstract

The Korea Atomic Energy Research Institute (KAERI) obtains UO_2_ powder using the ammonium uranyl carbonate (AUC) wet process. Hydrated lime (Ca(OH)_2_) is used to neutralize liquid wastes produced from the AUC process, and the resulting byproduct is known as lime waste. The purpose of this study is to determine optimum operating conditions for cementation of radioactive lime waste produced from the AUC process, and to evaluate the structural stability and leaching stability of cement waste form. The waste acceptance criteria (WAC) of a waste disposal facility in Korea were used to evaluate the cement waste form samples. The maximum lime waste content guaranteeing the shape stability of cement waste form was found to be 80 wt.% or less. Considering the economic feasibility and error of the cementation process, the optimum operating conditions were achieved at a lime waste content of 75 wt.% and a water-to-cement (w/c) ratio of 2.0. The compressive strength of cement waste form samples prepared under optimal operating conditions was 61.4, 76.3, and 61.0 kgf/cm^2^ after the thermal cycling test, water immersion test, and irradiation, respectively, satisfying the compressive strength of 35.2 kgf/cm^2^ specified in WAC. A leaching test was performed on the samples, and the leachability indexes (*LX*) of Cs, Sr, and Co nuclides were 7.63, 8.02, and 10.89, respectively, which are all higher than the acceptance criterion of 6. The results showed that the cement waste forms prepared under optimal operating conditions satisfied the WAC in terms of structural stability and leaching stability. As such, the proposed cement solidification method for lime waste disposal can be effective in solidifying lime waste powder produced during the neutralization of liquid wastes in the AUC process.

## 1. Introduction

The operation and decommissioning of nuclear power plants result in the production of radioactive waste. The uranium conversion facility of the Korea Atomic Energy Research Institute (KAERI) relies on the ammonium uranyl carbonate (AUC) wet process (Figure 1) to obtain UO_2_ powder. The AUC process is a uranium reconversion process involving (NH_4_)_4_UO_2_(CO_3_)_3_ and large amounts of nitric acid (HNO_3_). Hydrated lime (Ca(OH)_2_) is used to neutralize liquid wastes produced from the AUC process [1,2], and the resulting byproduct is known as lime waste.

As shown in Figure 1, this study examines the processing and disposal of a filter cake, which is radioactive lime waste in solid form. Lime waste, no longer produced as the uranium conversion facility has been decommissioned, is being stored in 435 drums (200 L/drum). While past research has explored methods of processing waste remaining after uranium recovery as non-radioactive industrial waste, such methods were deemed unfeasible due to the low rate of uranium recovery and production of secondary waste. Adequate procedures should be in place for stable disposal of lime waste produced from the AUC process, and lime waste has to undergo cementation in order to fulfill the waste acceptance criteria (WAC). The physical and chemical characteristics of cement waste form must also satisfy the WAC [3].

The cementation of waste should be a simple process with affordable installation costs. The technology employed should have little room for mistakes by operators or mechanical errors, and involve minimal increase in volume to keep disposal costs down. Various cementation processes with different solidifying mediums have been developed for cement, asphalt, polymer, and glass. Among them, the cement solidification technique is the best proven [4]. The advantages of cement solidification include the use of well-known materials and technology, a wide range of applications, and reasonable cost [5]. Thermo-hydro-mechanical mechanisms and thermodynamic studies related to cementation have been actively reported [6,7]. When cement is mixed with water, hydration and hydrolysis occur simultaneously or in succession, and the chemical reactions result in hardening [8]. The mixture maintains a workable state as a fluid or plastic compound for 1–2 h, and then begins to harden. As such, it must be uniformly mixed and placed in a suitable container before the hardening process [9].

Portland cement (type I) has been used as a solidifying medium for a long time, and its technical safety and convenience have already been proven [10]. The initial setting time of this solidifying medium is approximately four hours, which secures sufficient time for uniform mixing with materials of the cementation process [11]. Alumina cement is characterized by fast condensation and high exothermic nature [12,13]. Condensation is delayed by adding slowing agents, and pores are removed using antifoaming agents, flow improvers, and enhancing agents [14,15]. The process becomes more complex with the introduction of additives, which cause quality to be influenced by more variables. Most additives are surfactants that release gas under irradiation, and must be prohibited for safety reasons in disposal facilities.

The purpose of this study is to determine optimum operating conditions of cementation of radioactive lime waste produced from the AUC process, and to evaluate the structural stability and leaching stability of cement waste form. The WAC of a waste disposal facility in Korea was used to evaluate the cement waste form samples.

## 2. Materials and Methods

### 2.1. Materials

The radioactive lime waste used in the experiment is lime waste produced from KAERI’s AUC process; its characteristics are presented in Table 1. U-235 and Cs-137, with concentrations of 55.4 Bq/g and below 0.200 Bq/g, respectively, are classified as low-level radioactive wastes.

The cement used for cementation was Portland cement (type I), and the radioactive lime waste used in the experiment was dried at 105 °C for 24 h. Moisture was removed from lime waste because the water-to-cement (w/c) ratio is an important variable. To prepare uniform cement waste form samples, dried lime waste was pulverized into powder using a mortar. Lime waste is easily pulverized, and disperses well in water. Figure 2 shows the dried and powdered lime waste used in the experiment.

### 2.2. Experimental Methods and Evaluation

#### 2.2.1. Maximum Content of Lime Waste

The shape stability of lime waste and cement compound was evaluated with varying lime waste contents of 71, 75, and 80 wt.% in cement waste form samples.

#### 2.2.2. Mixing Ratio of Cement Waste Form

The physical and chemical characteristics of cement waste form depend on the water-to-cement (w/c) ratio. Mixing is impossible with a small w/c ratio, meaning that cement waste form cannot be produced. In the case of mixtures having poor flowability, air gets trapped during the mixing process, causing the cement waste form to be highly porous. The pores serve as leaching paths, and contribute to high infiltration of water and leaching of nuclides. In the experiment, the weight ratio of lime waste was fixed at 75 wt.%, and varying w/c ratios of 1.55, 1.65, 1.75, 1.85, 1.95, 2.00, 2.20, and 2.40 were used to determine the optimal mixing ratio.

#### 2.2.3. Workability Test

Cement mixtures must have flowability to be usable in cementation. Since flowability is highly influenced by the amount of water added, the w/c ratio is an important variable. The workability test determines the range of w/c ratios that allows mixtures to have appropriate flowability for cementation. The weight ratio of lime waste was fixed at 75 wt.%, and observations were made with varying w/c ratios of 1.55, 1.65, 1.75, 1.85, 1.95, 2.00, 2.20, and 2.40 (same as the mixing ratios of cement waste form in Section 2.2.2).

#### 2.2.4. Preparation and Curing of Cement Waste Form

Cement waste form and lime waste were mixed in accordance with “Testing Method for Mechanical Mixing of Hydraulic Cement Pastes and Mortars of Plastic Consistency” (KS-L-5109, South Korea) using a mechanical mixer (JI-206, JEIL, Seoul, Korea) [16]. The cement mixtures were poured into polyethylene molds (D = 5 cm, H = 12 cm) at room temperature (18–25 °C) and a relative humidity range of 30–60%. The molds were covered with lids to prevent moisture from evaporating, and then cured for 28 days. The amount of free standing water remaining in the molds after 28 days was measured using a graduated pipette. The molds were carefully removed to prevent damage to the samples, which were fabricated to have a diameter of 5 cm and height of 10 cm. Figure 3 shows the curing of samples prepared using lime waste and cement.

#### 2.2.5. Cement Waste Form Property Tests

The cement waste form samples prepared according to the preparation and curing method described above (Section 2.2.4) were tested for structural stability and leaching stability. WAC was employed as the evaluation method and criteria. The disposal facility in Korea is being operated by Korea Radioactive Waste Agency (KORAD); Table 2 presents the test items and methods in KORAD’s WAC [3]. Three samples were prepared for each item, and the average values for the three samples were used in the evaluation.

The leaching test was performed as an inactive test, with CsCl, CoCl_2_6H_2_O, and SrCl_2_6H_2_O as reagents. The reagents were completely dissolved in water before use, and Cs, Co, and Sr each had a concentration of 2000 ppm.

The leachability of cement waste form was evaluated based on the leachability Index (*LX*). Here, *LX* is defined as follows.
(1)$$LX=-logD_{e}$$
*D_e_* = effective diffusion coefficient [cm^2^/sec]
(2)$$CFL=f\left(t\right)=\frac{\sum a_{n}}{A_{0}}=2\frac{S}{V}{\left[\frac{D_{e}}{\pi }\right]}^{1/2}\cdot t^{1/2}\;$$

Based on Equation (2), the plot of $\left[\frac{\sum a_{n}}{A_{0}}\right]$ with respect to $\left[\sqrt{t}\right]$ has the following slope:$$\mathrm{Slope}=2\frac{S}{V}{\left[\frac{D_{e}}{\pi }\right]}^{1/2}$$

By substituting slope into Equation (3), *D_e_* can be calculated as follows.
(3)$$D_{e}={\left(slope\right)}^{2}\cdot \left[\frac{1}{2}\frac{V}{S}\right]\cdot \pi$$

*CFL* = cumulative fraction leached; an = the total amount of material released during leaching periods up to time t; *A*_0_ = the initial amount of material; *V* = waste form volume; *S* = the surface area of waste form.

## 3. Results

### 3.1. Characteristics of Lime Waste

Lime waste is presumed to be composed of chemical substances used in the AUC process, namely, NH_4_NO_3_, NaNO_3_, Ca(NO_3_)_2_, CaCO_3_, and Ca(OH)_2_ [18]. To determine the crystalline structure of lime waste used in sample preparation, X-ray diffraction analysis (XRD) was performed with an X-ray diffractometer (D8 DISCOVER, Bruker, Billerica, MA, USA). Figure 4 shows the XRD patterns of lime waste. The crystalline structure of lime waste was found to be comprised of CaCO_3_ and Ca(NO_3_)_2_ [19,20,21].

### 3.2. Operating Range and Optimum Conditions

The total volume of cement waste form to be disposed decreases with increasing lime waste content, mixed with cement when preparing the samples. However, the weight ratios of lime waste and cement influence the physical stability of cement waste form, which contains lime waste [22]. This study determined the maximum lime waste content guaranteeing the physical stability of cement waste form.

The shape stability of lime waste and cement mixtures was evaluated with varying lime waste content of 71, 75, and 80 wt.%. Figure 5 shows the shape stability of cement waste form with varying lime waste content of 71, 75, and 80 wt.%. At a lime waste content of 80 wt.%, the samples collapsed when the polyethylene molds were removed after curing (Figure 5c). The maximum lime waste content was evaluated to be below 80 wt.%. Considering the economic feasibility and stability of cement waste form, the weight ratio of lime waste introduced in preparing the samples was set as 75 wt.%.

The workability of cement–lime mixtures was evaluated in relation to a w/c ratio with the lime waste content fixed at 75 wt.%, and the results are shown in Figure 6. The mixtures had no flowability at a w/c of 1.55, 1.65, and 1.75, but exhibited good flowability at a w/c of 1.95 or higher.

The samples were observed for cracks and collapse while performing the workability test, free standing water test, and water immersion test with varying w/c ratios. These observations were used to determine the operating range for the preparation of cement waste form samples, and the results are shown in Table 3. Cement waste form cannot be prepared at a w/c ratio of 1.55–1.75, and the range of w/c ratios with acceptable mixability and flowability was 1.85–2.40. However, free standing water was produced at a w/c ratio above 2.40, and samples exceeding the criterion of 0.5 vol.% were deemed inadequate. In the water immersion test, all samples prepared at a w/c ratio of 1.85–2.40 had no cracks or collapses. The range of w/c ratios satisfying all requirements in terms of workability, free standing water, and water immersion was 1.95–2.20. When the lime waste content was 75 wt.%, the corresponding range of w/c ratios was 1.90–2.30.

Figure 7 shows the operating range of the cementation process in relation to the w/c ratio of lime waste when the lime waste content is 75 wt.%, meaning that base cement content is 25 wt.%. Based on the flowability test and free standing water limit, the range of w/c ratio compatible with the preparation of cement waste form was found to be 1.90–2.30. Testing was not required at a w/c ratio of 2.40 because it was excluded from the operating range in the free standing water test guidelines of KORAD’s WAC.

Taking into account variables such as the scale up factor of the process, equipment precision, and expertise of operators, a w/c ratio of 2.0 was used to prepare the cement waste form samples. Considering the economic feasibility and error of the cementation process, the weight ratio of lime waste was set as 75 wt.% and that of Portland cement (type I) as 25 wt.%. Table 4 shows the components and ratios of cement waste form samples. The samples had a density of 1.445 g/cm^3^, which translates to a 15% increase in volume since cement has a density of 3.16 g/cm^3^.

### 3.3. Evaluation of Disposal Feasibility

#### 3.3.1. Structural Stability of Cement Waste Form

The disposal feasibility was evaluated with cement waste form samples cured at a w/c ratio of 2.0. The evaluation was comprised of a leaching test, thermal cycling test, irradiation test, water immersion test, and compressive strength test.

Figure 8 shows the changes in volume and weight of the samples before and after the thermal cycling test, irradiation test, and water immersion test. The weight of samples decreased by 16.25%, 15.00%, and 7.33%, respectively, after the thermal cycling test, irradiation test, and water immersion test. The changes in volume were within 0.3% after each test. Figure 9 shows the compressive strength of samples before and after the thermal cycling test, irradiation test, and water immersion test. The actual photos of samples representing changes in compressive strength and the setup of the thermal cycling test are presented in Figure 10.

The initial compressive strength of cement waste form samples after curing was 55.0 kgf/cm^2^, higher than the acceptance criterion of 35.2 kgf/cm^2^. The compressive strength was 61.4, 76.3, and 61.0 kgf/cm^2^ after the thermal cycling test, water immersion test, and irradiation, respectively, thus satisfying the criterion of the disposal facility.

#### 3.3.2. Leaching Stability of Cement Waste Form

The leaching test was performed on cement waste form samples for 90 days in accordance with ANS 16.1. Table 5 shows De calculated for Cs, Sr, and Co nuclides in cement waste form and the concentration of leached Co. In the leaching test, the concentration of Co was ND (non-detectable, <0.05) for 19 days.

The leaching of nuclides is more consistent with the typical diffusion model if the cumulative leached fraction [∑a_n_/A_o_] in relation to leaching time [(day)^1/2^] is closer to a linear distribution [19]. Figure 11 shows the cumulative leached fraction [∑a_n_/A_o_] of Cs, Sr, and Co in relation to leaching time [(day)^1/2^] based on the leaching test. The nuclides arranged in the order of slope were Cs > Sr > Co. This means that Cs has a faster leaching rate than Sr and Co. The slope of Co was not reliable as most leaching concentration values were ND.

The slopes representing *D_e_* were substituted into Equation (1) to calculate the *LX*. The cement waste form samples must have a leachability index above 6 to ensure disposal feasibility (Table 2). Table 6 shows *D_e_* and *LX* of Cs, Sr, and Co nuclides. The average *LX* of Cs, Sr, and Co was 7.63, 8.02, and 10.89, respectively, with one sample for Co exceeding 11. Figure 12 shows the average *LX* of Cs, Sr, and Co nuclides. The *LX* of Cs, Sr, and Co nuclides satisfied the waste acceptance criteria of the disposal facility, thereby verifying their disposal feasibility.

## 4. Conclusions

This study determined the operating range and optimum operating conditions for the cementation of radioactive lime waste produced from the AUC process. The maximum lime waste content guaranteeing the shape stability of cement waste form was evaluated to be below 80 wt.%. When the weight ratio of lime waste was 75 wt.%, the range of w/c ratios for cement solidification was 1.90 to 2.30. Considering the economic feasibility and error of the cementation process, the lime waste content was set as 75 wt.% and the w/c ratio as 2.0. These were found to be optimum operating conditions for the cementation of lime waste.

The cement waste form samples prepared under optimum operating conditions were tested for structural stability and leaching stability. The disposal feasibility of samples was examined based on the compressive strength and leaching test procedures of WAC, which was employed as the evaluation method and criteria. The samples satisfied the WAC criteria in terms of initial compressive strength after curing, compressive strength after the thermal cycling test, compressive strength after the water immersion test, compressive strength after irradiation, and leachability of Cs, Sr, and Co.

As such, the proposed cement solidification method for lime waste disposal can be effective in solidifying lime waste powder produced during the neutralization of liquid wastes in the AUC process.

## Figures and Tables

**Figure 1 materials-15-00872-f001:**
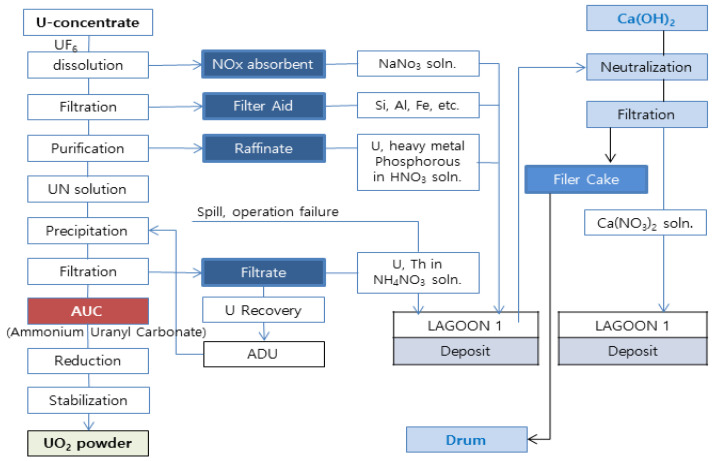
Schematic diagram of reconversion process and waste.

**Figure 2 materials-15-00872-f002:**
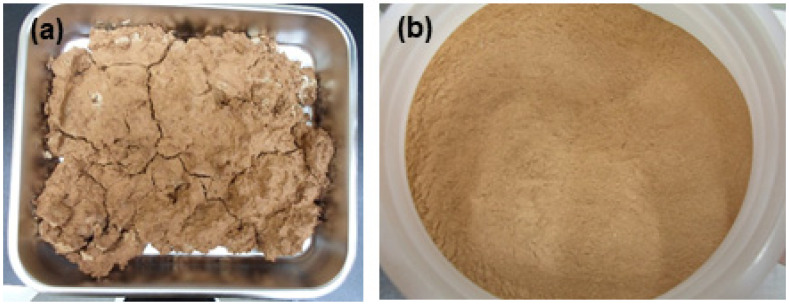
Lime waste used in the experiment is (**a**) dried and (**b**) powdered.

**Figure 3 materials-15-00872-f003:**
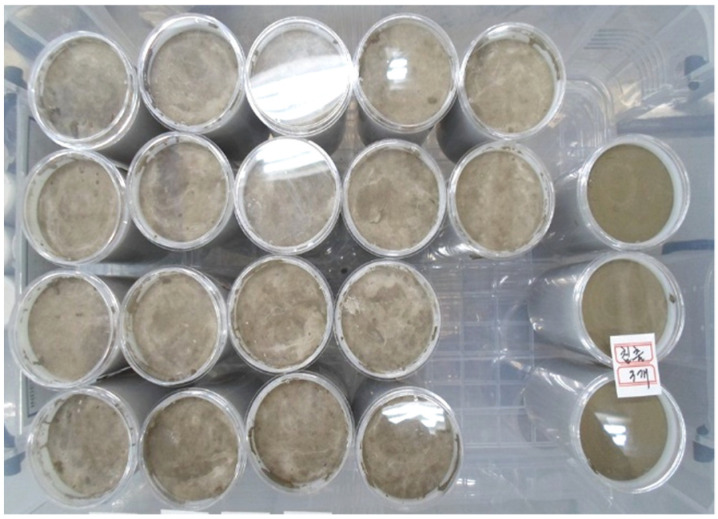
Curing of specimen that blocked movement of moisture with lid closed.

**Figure 4 materials-15-00872-f004:**
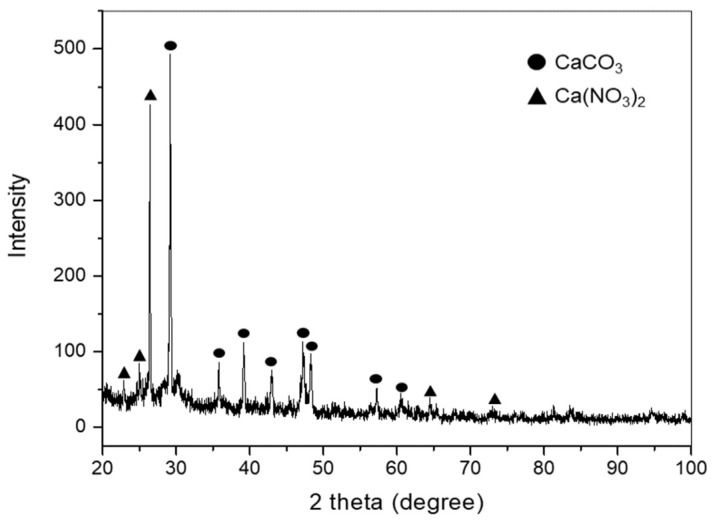
Results of XRD analysis of lime waste.

**Figure 5 materials-15-00872-f005:**
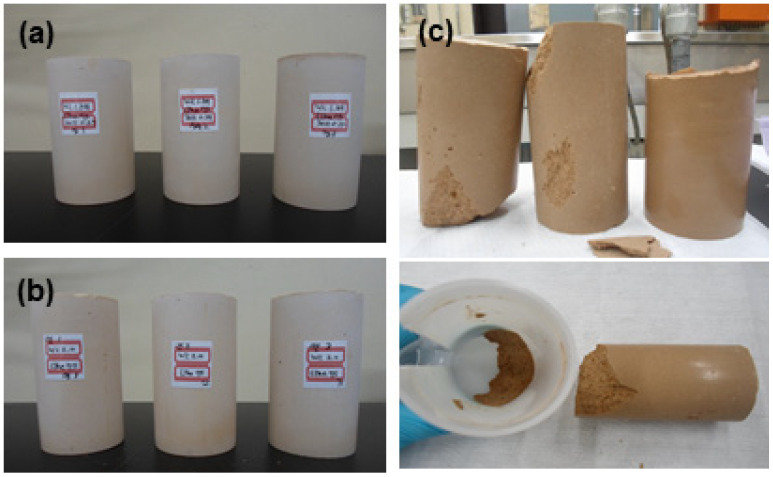
Shape stability of cement waste forms with respect to lime waste content of (**a**) 71 wt.%, (**b**) 75 wt.%, and (**c**) 80 wt.%.

**Figure 6 materials-15-00872-f006:**
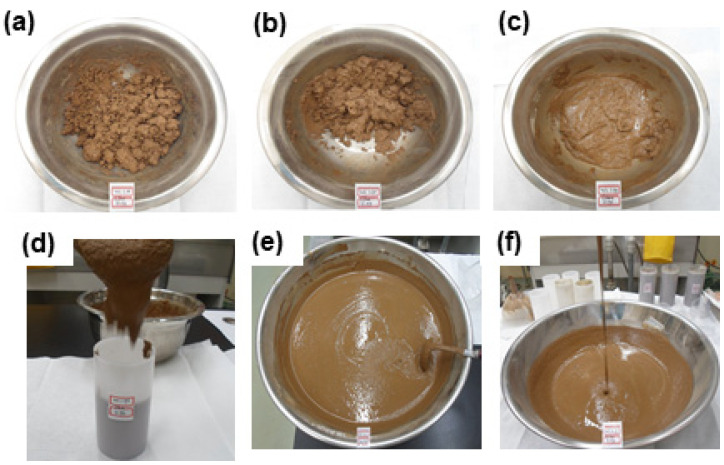
Workability test results of 75 wt.% of lime waste and cement mixture according to w/c ratios of (**a**) 1.55, (**b**) 1.65, (**c**) 1.75, (**d**) 1.95, (**e**) 2.00, and (**f**) 2.20.

**Figure 7 materials-15-00872-f007:**
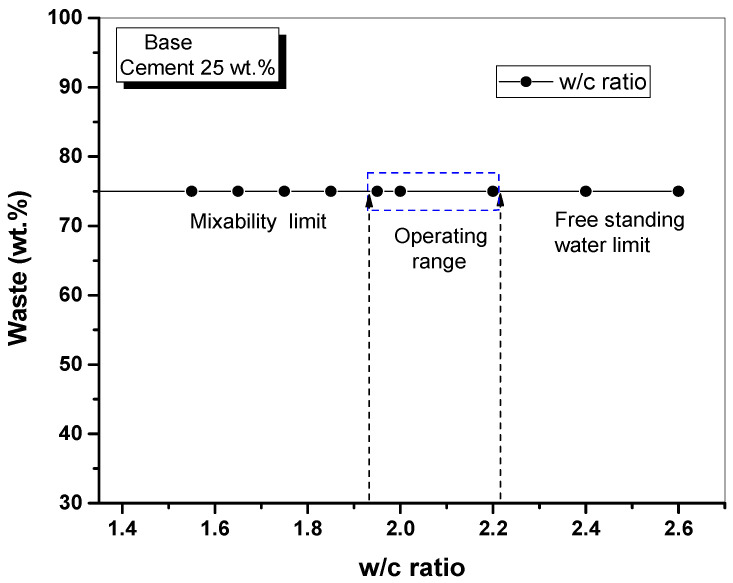
Operating range of cementation process according to w/c ratio of lime waste.

**Figure 8 materials-15-00872-f008:**
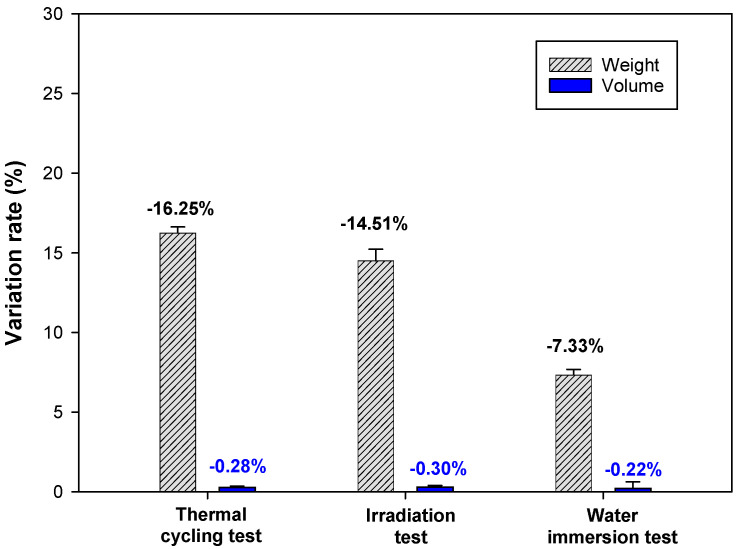
Changes in volume and weight before and after thermal cycling test, irradiation test, and water immersion test (w/c 2.0, lime waste 75 wt.%).

**Figure 9 materials-15-00872-f009:**
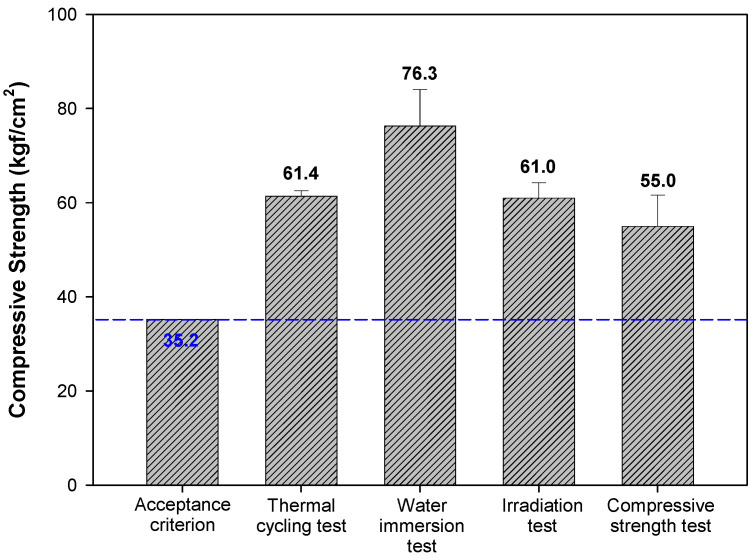
Evaluation of acceptance criteria of the final disposal facility (w/c 2.0, lime waste 75 wt.%).

**Figure 10 materials-15-00872-f010:**
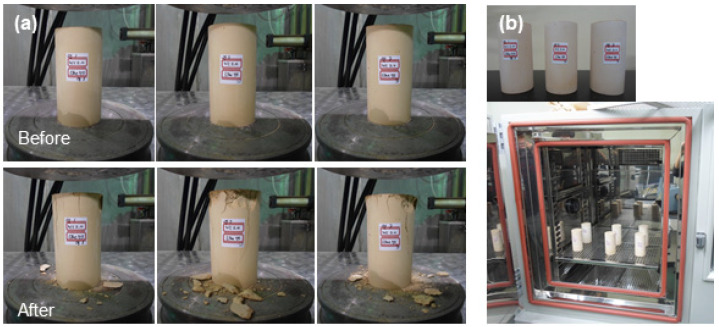
Appearance of compressive strength and thermal cycling test. (**a**) Compressive strength, (**b**) thermal cycling test.

**Figure 11 materials-15-00872-f011:**
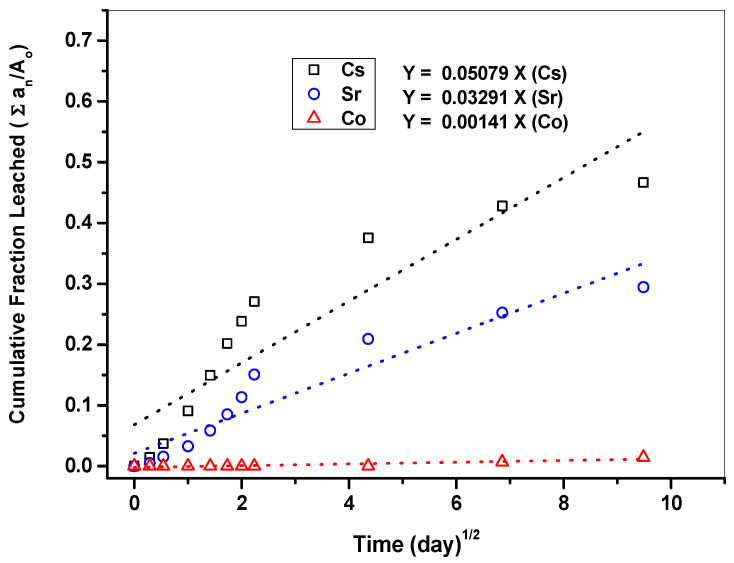
Cumulative fraction of Cs, Sr, Co leaching from cement waste form (w/c 2.0, Lime waste 75 wt.%).

**Figure 12 materials-15-00872-f012:**
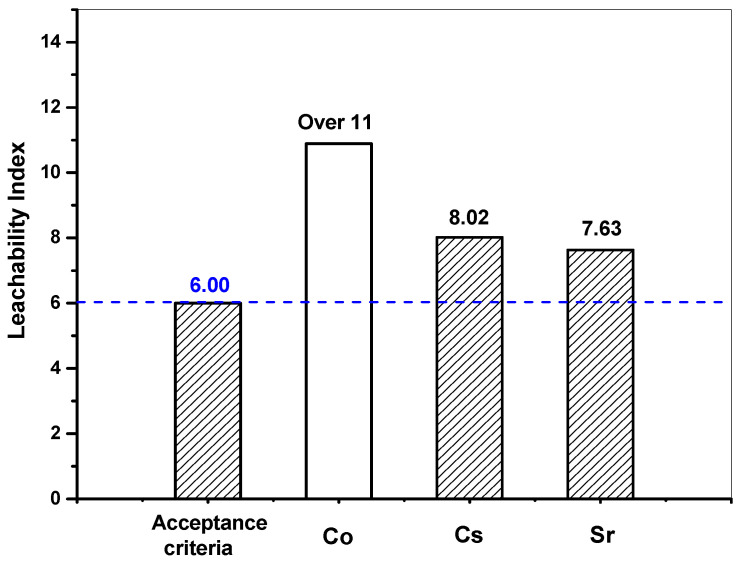
Leachability index of respective nuclides included in cement waste form (w/c 2.0, lime waste 75 wt.%).

**Table 1 materials-15-00872-t001:** Characteristics of lime waste being stored.

Storage Volume [200 L drum]	Radioactivity [Bq/g]	Components	pH
435	U-235: 55.4 Cs-137 < 0.200	CaCO_3_ Ca(NO_3_)_2_	8.2

**Table 2 materials-15-00872-t002:** Test items and methods specified in KORAD acquisition criteria [17].

Item	Test	Standard Method	Test Method	Criteria
Structural stability	Compressive strength test	KS F2405	-	≥35.2 kgf/cm^2^ (3.44 Mpa)
	Water immersion test (90 days)	NRC ^1^	Compressive strength after immersion test	≥35.2 kgf/cm^2^
	Thermal cycling test (28 days)	ASTM B553	Compressive strength after thermal cycling test	≥35.2 kgf/cm^2^
	Irradiation test	NRC ^1^	Compressive strength after irradiation (1.0 × 10^7^ Gy)	≥35.2 kgf/cm^2^
Leachability	Leaching test (90 days)	ANS 16.1	Cs, Sr, Co	Leachability index ≥ 6
Free standing water test	Sample	EPA ^2^	-	>0.5 vol.%
	200 L drum	ANS 55.1	-	>0.5 vol.%

^1^ NRC, Waste Form Technical Position, Revision 1. (1991). ^2^ EPA, Method 9095B “Paint Filter Liquids Test”.

**Table 3 materials-15-00872-t003:** Operating range according to w/c ratio, workability, and free standing water of lime waste.

Ratio		Workability	No. Sample	Free Standing Water		Water Immersion Test
w/c	Waste (wt.%)			Measurement (mL)	Criteria (≤0.5 vol/%)	No Cracks/ Collapses
1.55	75	No workability				
1.65	75	No workability				
1.75	75	No workability				
1.85	75	Bad	1	0.0	Less than 0.5 vol.%	o
			2	0.0		o
			3	0.0		o
1.95	75	Good	1	0.0		o
			2	0.0		o
			3	0.0		o
2.00	75	Very good	1	0.0		o
			2	0.0		o
			3	0.0		o
2.20	75	Very good	1	0.8		o
			2	0.0		o
			3	0.0		o
2.40	75	Very good	1	2.6	More than 0.5 vol.%	o
			2	3.1		o
			3	2.5		o

**Table 4 materials-15-00872-t004:** W/c and waste content of cement waste forms.

Cement	Ratio (Weight)			Cement Waste Form
Portland cement type I	w/c	Lime/ (lime + cement)	Cement/ (lime + cement)	Density [g/cm^3^]
	2.00	0.75	0.25	1.445

**Table 5 materials-15-00872-t005:** Cumulative leached fraction of nuclides (Cs, Sr, Co) in cement waste form.

No.	Σ(Δt)n (Day)	Σ(Δt)n (Day)^1/2^	Cumulative Leached Fraction [∑a_n_/A_o_]			Co Concentration, a_n_ (mg/L)
			Cs	Sr	Co	
1	0.1	0.29	0.01	0.01	0.000	<0.05
2	0.3	0.54	0.04	0.02	0.000	<0.05
3	1	1.00	0.09	0.03	0.000	<0.05
4	2	1.41	0.15	0.06	0.000	<0.05
5	3	1.73	0.20	0.09	0.000	<0.05
6	4	2.00	0.24	0.11	0.000	<0.05
7	5	2.24	0.27	0.15	0.000	<0.05
8	19	4.36	0.38	0.21	0.000	<0.05
9	47	6.86	0.43	0.25	0.007	1.53
10	90	9.49	0.47	0.30	0.015	1.60

**Table 6 materials-15-00872-t006:** Effective diffusion coefficient (*D_e_*) and leachability index (*LX*).

Nuclides	Sample No.	V/S (cm)	Slope	*D_e_* (cm^2^/day)	*D_e_* (cm^2^/sec)	*LX*
Co	S1	0.999	4.87 × 10^−4^	1.86 × 10^−7^	2.15 × 10^−12^	11.67
	S2	1.000	1.89 × 10^−3^	2.80 × 10^−6^	3.25 × 10^−11^	10.49
	S3	1.000	1.86 × 10^−3^	2.72 × 10^−6^	3.14 × 10^−11^	10.50
	Average	1.000	1.41 × 10^−3^	1.90 × 10^−6^	2.20 × 10^−11^	10.89
Sr	S1	0.999	3.14 × 10^−2^	7.74 × 10^−4^	8.96 × 10^−9^	8.05
	S2	1.000	3.43 × 10^−2^	9.24 × 10^−4^	1.07 × 10^−8^	7.97
	S3	1.000	3.22 × 10^−2^	8.12 × 10^−4^	9.40 × 10^−9^	8.03
	Average	1.000	3.26 × 10^−2^	8.37 × 10^−4^	9.68 × 10^−9^	8.02
Cs	S1	0.999	5.05 × 10^−2^	2.00 × 10^−3^	2.31 × 10^−8^	7.64
	S2	1.000	5.13 × 10^−2^	2.06 × 10^−3^	2.39 × 10^−8^	7.62
	S3	1.000	5.06 × 10^−2^	2.01 × 10^−3^	2.33 × 10^−8^	7.63
	Average	1.000	5.08 × 10^−2^	2.02 × 10^−3^	2.34 × 10^−8^	7.63

## Data Availability

The data presented in this study are available on request from the corresponding author. The data are not publicly available due to institutional and national data sharing restrictions.

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
