# Peer review of "Evaluation of Disposal Stability for Cement Solidification of Lime Waste"

_materials, 2022, doi:10.3390/ma15030872_

Round 1

Reviewer 1 Report

This study demonstrates some practical significance for engineering application if its performance with good stability and mechanical strength.

The author needs to proceed some stability study with a systematical and logical experimental design.

the overall writing should be improved both in english and logicla manner to make it concise and easy understanding.

Proper references should be cited, but those un-related citations should be removed. 

Author Response

We have attached the answer to your comments as a file in Word format.

Reviewer 2 Report

Reviewer:

The purpose of this manuscript is to determine optimum operating conditions for cementation of radioactive lime waste produced from the AUC process. This manuscript evaluated the structural stability and leaching stability of cement solidification wastes. Research background is of significance; objectives are also valuable. The scientific content is of relevance to the reader of the journal. However, the paper has the following problems, which needs a major improvement.

  1. In introduction: There are some important discussions about the solidification and leaching stability of cement waste. This is very meaningful. However, since many factors (e.g., temperature effect, multi physical field process, etc.) is involved in the radioactive waste, the recent relevant international achievements must be mentioned. For example, “Coupled thermo-hydro-mechanical mechanism in view of the soil particle rearrangement of granular thermodynamics. Computers and Geotechnics, 2021,137(8),104272”, “Temperature-driven migration of heavy metal Pb2+ along with moisture movement in unsaturated soils. International Journal of Heat and Mass Transfer, 2020,153: 119573”, and “A thermodynamic constitutive model with temperature effect based on particle rearrangement for geomaterials. Mechanics of Materials, 2019, 139: 103180”. This is of forward-looking significance for the further development of this paper.
  2. Line 85: The chemical composition of the used cement needs to be clearly given. This is closely related to the curing effect of cement solidification.
  3. Line 121: Generally, temperature has a great influence on the test results as the listed above references. In the manuscript, the temperature is in the range of 18°C-25°C. How much will this temperature range affect the test results? Need an explanation?
  4. Lines 147: What is the meaning of the definition of LX expressed in logarithm? Appropriate instructions are required.
  5. Lines 150: What is the meaning of f(t). Since f(t) already exists, why is "CFL" defined?
  6. Line 168: Figure 4 is not clear and needs to be improved. The points in Fig. 4 need to be identified. What are their meanings? Need explanation?
  7. Line 243: The unit of longitudinal compressive strength in Fig. 9 should be in SI units?
  8. Figs 8 and 9: Can the deviation of test results be given in Figures 8 and 9? Please make appropriate supplement or discussion.

Finally, the manuscript must be subject to the above modifications before publication.

Author Response

(The authors gave the same response as above.)

Reviewer 3 Report

The submitted paper " Evaluation of disposal stability for cement solidification of lime waste" is addressing an important and interesting topic, therefore thank you very much for your work and the contribution.
The aim of the paper is to determine optimum operating conditions for cementation of radioactive lime waste produced from the AUC process, and to evaluate the structural stability and leaching stability of cement waste form. In my opinion the topics addressed in the paper, from a practical and methodological point of views, are up to date and current. 

The paper has a very clear and in my opinion valuable practical contribution. The manuscript is written according to the rules and reads smoothly. 
My dissatisfaction - despite well constructed and presented models - leaves the fomal scientific part of the paper.
I would suggest to the authors to clearly strip the manuscript of its case study character and give it a stronger scientific character. A good solution would be to present the current state of art/review of mathematical methods used in modeling similar processes and at the same time to justify the model used. This can be done in the introduction section or as a separate literature review section.

Author Response

(The authors gave the same response as above.)

Round 2

Reviewer 2 Report

The authors revised the manuscript according to the requirements. Recommended for publication.